# Three-dimensional beating pattern of the ciliary tip in the live ciliate *Tetrahymena*

Akisato Marumo[1,*], Hiroto Ishii[1,*], Shin Yamaguchi[1,*], Rieko Sumiyoshi[1], Kyohei Matsuda[1], Masahiko Yamagishi[1,2] and Junichiro Yajima[1,2,3,‡]

## ABSTRACT

Ciliates utilise motile cilia, which are highly dynamic organelles protruding from the cell surface, to swim helically in a three-dimensional (3D) space. The 3D nature of their swimming behaviour and rapid ciliary beatings make its quantitative analysis difficult. Here, we quantified the 3D motion of a microbead bound to a ciliary tip in a live immobilised *Tetrahymena thermophila* cell using 3D tracking optical microscopy. We found that the tip of individual ciliate cilia, consisting of the 9+2 structure of the axoneme, shows semicircular counterclockwise rotation in a single plane when looking down on the cilium. The rotational trajectories of the tip consist of fast and slow strokes, with the tip path during the fast and slow strokes being an arc and linear, respectively. The direction of the fast stroke of the ciliary tip, with respect to the cell body, was from the right-anterior to the left-posterior region, which is consistent with the direction that would induce right-handed helical swimming of the *Tetrahymena*.

KEY WORDS: Cilia, Tetrahymena, Ciliate, 3D tracking observation

## INTRODUCTION

Ciliates, such as *Tetrahymena*, are eukaryotic unicellular microorganisms that swim along helical paths within a three-dimensional (3D) space (Blake and Sleigh, 1974; Machemer, 1972). Helical swimming is an effective method of propulsion for ciliates living in an environment of low Reynolds number (Blake and Sleigh, 1974; Jennings, 1901; Purcell, 1977). Such helical ciliate swimming is propelled by the periodic beating motion of microtubule-based cellular organelles called motile cilia (∼5–10 µm long and 0.3 µm in diameter; Jiang et al., 2015; Reynolds et al., 2018) that protrude from the cell surface along ciliary rows. Motile cilia have a conserved structure called an axoneme, a tube-shaped structure consisting of nine outer doublet microtubules surrounding a pair of central microtubules, known as a 9+2 microtubule arrangement. The ensemble action of axonemal dynein ATPase motor proteins placed between adjacent doublet microtubules in the axoneme cause a well-ordered beating motion of the cilia (Lin and Nicastro, 2018; Satir, 1968).

The beating motion of live ciliate cilia has been captured on one focal plane using high-speed microcinematography (Baba and Hiramoto, 1970) and high-speed cameras (Soh et al., 2022; Stoddard et al., 2018; Wood et al., 2007), which revealed that the cilia beat in a periodic and asymmetric manner. The beating of individual cilia consists of two distinct phases – an effective stroke that beats quickly backward along a path of high drag away from the cell surface, which generates a propulsive force, and a recovery stroke that returns relatively slowly along a path of low drag near the cell surface to the starting point of the next effective stroke in a bent shape like a whip motion (Machemer, 1972; Naitoh and Sugino, 1984; Omoto and Kung, 1980). The asymmetry of the beat pattern results in net fluid flow in the direction of the effective stroke, which might be responsible for the unidirectional helical swimming of the cell (Machemer, 1972; Marumo et al., 2021). High-speed two-dimensional (2D) imaging (2000 frames s$^{-1}$) of live *Tetrahymena* cilia beating from the side view has revealed that cilia are relatively straight in shape during the effective stroke (Stoddard et al., 2018). A recent excellent method of immobilizing the live *Tetrahymena* cell allows the cilia to be imaged from above the cell, consequently enabling quantification of the relative duration of the effective stroke as well as the recovery stroke of different cilia (Soh et al., 2022). However, given that ciliary motion occurs in three dimensions, which is not limited to only one focal plane, information on 2D projection through classical microscopy is often unsatisfactory for quantifying ciliary beating.

Motile cilia in various types of cells and tissues are essential for cell locomotion and fluid flow over the epithelium. Motile cilia with a 9+2 configuration of microtubules also exist on the tracheal epithelial and brain ependymal cell surfaces, which are responsible for foreign body exclusion in the trachea (Fliegauf et al., 2007) and brain development (Park et al., 2019), respectively, and show a planar back-and-forth beating pattern through a slightly different path (Katoh et al., 2018; Ueno et al., 2012). By contrast, the cone-like 3D beating pattern of specialised ciliary assemblies, consisting of ∼75 cilia that are up to 55 µm long, called cirrus in hypotrichs ciliate *Stylonychia mytilus* has been estimated by analysing sequential data obtained using the anaxial illumination method for simultaneous records of stereoscopic video image (Teunis and Machemer, 1994). Other motile cilia with 9+0 configuration of microtubules are found in the embryonic node, which is responsible for establishing left-right asymmetry of visceral organs, and such node cilia beat with a rotary pattern (Nonaka et al., 1998). 3D

[1]Department of Life Sciences, Graduate School of Arts and Sciences, The University of Tokyo, 3-8-1 Komaba, Meguro-ku, Tokyo 153-8902, Japan. [2]Komaba Institute for Science, The University of Tokyo, 3-8-1 Komaba, Meguro-ku, Tokyo 153-8902, Japan. [3]Research Center for Complex Systems Biology, Universal Biology Institute, The University of Tokyo, 3-8-1 Komaba, Meguro-ku, Tokyo 153-8902, Japan.
*These authors contributed equally to this work

‡Author for correspondence (yajima@g.ecc.u-tokyo.ac.jp)

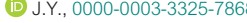 J.Y., 0000-0003-3325-7868

quantification of individual ciliary motion would be helpful for better understanding how the ciliary motion of motile cilia contributes to force generation.

Recently, we reported that *Tetrahymena* swims along a right-handed helical trajectory at a high speed of ~400 μm s$^{-1}$, accompanied by right-handed rolling of its cell body in free space (Marumo et al., 2021). This finding was obtained through 3D tracking of *Tetrahymena* cells that had internalised fluorescent microbeads via phagocytosis, using a three-dimensional prismatic optical tracking (tPOT) microscope originally developed by our group (Yajima et al., 2008). Owing to the dense protrusion of numerous cilia from the surface of *Tetrahymena* (Bayless et al., 2019), which swims rapidly in a helical manner (Ishikawa and Kikuchi, 2018; Marumo et al., 2021), it is difficult to observe the 3D motion of individual cilia in live *Tetrahymena*. In this study, we tracked the 3D trajectories of fluorescent microbeads attached to the tips of cilia in live immobilised *Tetrahymena thermophila*. This was achieved by combining the tPOT microscope, which provides positional information in 3D (Yajima et al., 2008), with micropipette manipulation (Iwadate, 2003) to aspirate and trap a single live *Tetrahymena*. We reported that the tip of an individual cilium beats in a counter-clockwise (CCW) rotary pattern consisting of the fast motion (i.e. effective stroke) and slow motion (i.e. recovery stroke). The direction of the fastest movement of the ciliary tip was from the front right to the back left, assuming that the cilia are observed from the tip to the base with the cell facing forward. Furthermore, we suggest the ciliary beating patterns underlying the formation of the helical swimming pattern of *Tetrahymena* in 3D space.

## RESULTS AND DISCUSSION
### Direct 3D observation near the tip of an individual cilium protruding from the surface of a live *Tetrahymena* cell

To gain insights into the beating patterns of a single cilium with an axoneme consisting of a 9+2 configuration of microtubules, we examined the 3D motion of a microbead attached near the tip of an individual cilium of live *Tetrahymena* cells. Surface proteins of cilia were biotinylated for fixation on neutravidin-coated fluorescent microbeads (0.2 μm in diameter) as probes for observation (Fig. S1), and the cells were immobilised by aspiration with a micropipette (Fig. 1A,B). Our experimental setup allows the 3D behaviour of the microbead to be recorded, reflecting the beating motion of an individual cilium. A portion of the cell was aspirated inside the micropipette, whereas most of the cilia, protruding from the cell surface, continued beating outside the pipette (Fig. 1B). When aspiration stopped, the cells began to swim freely (Movie 1), suggesting that micropipette aspiration did not significantly affect cell swimming. The 3D motion of the microbead bound to an individual cilium was monitored using a tPOT microscope (Yajima et al., 2008) (Fig. 1C). In this 3D tracking technique, the light beam flux from the sample is split into two light paths (blue and red in Fig. 1C) by a wedge prism. *Z*-directed movement (parallel to the beam flux) was estimated from the relative displacement of the two images produced by the split beams, whereas *x*- and *y*-directed movements (perpendicular to the beam flux) were determined from the average displacement of these images with nanometre accuracy (Maruyama et al., 2021) (Fig. S2). Using this method, we found that a microbead bound near the tip of an individual cilium of a live ciliate *Tetrahymena* mainly showed a rotary pattern, as seen in the superimposed image of 500 frames (1.14 s) (Fig. 1D,E). To quantify the 3D motion of a single cilium (Fig. 1F; Movie 2), the *xyz* trajectories of the microbead were rotated as viewed from above,

such that the tangential plane of the cell, where the ciliary base is presumed to be located, was aligned with the *xy*-plane, and the cell was oriented along the positive *y*-axis (Fig. 1G). The rotated 3D trajectories of the ciliary tip demonstrated that the direction of rotation was always CCW when viewed from tip to base (top view). This direction of rotation was consistent with that observed in 2D top-down imaging of cilia in magnetically immobilised iron-phagocytic *Tetrahymena* (Soh et al., 2022), as well as in cilia on the cortical sheet of *Paramecium,* another ciliate, extracted with Triton-glycerol (Noguchi et al., 1991).

Rotated *xyz* trajectories also revealed that the ciliary tip followed an orbital path as if drawing a semi-circle on almost the same *xyz* plane, and without such a tip movement in different planes in 3D space as might be expected if cilia beat with a 'complicated' whip-like motion (Fig. S3). The rotational trajectory driven by the ciliate cilia containing the 9+2 structure of the axoneme is reminiscent of the gyration of motile node cilia containing the 9+0 structure of the axoneme (Nonaka et al., 1998), the cirrus, consisting of specialised ciliary assemblies, in the ciliate *Stylonychia mytilus* (Teunis and Machemer, 1994), and most artificial cilia (Zhang et al., 2020). In contrast, the ciliate ciliary tip trajectory appears to be distinct from other motile cilia containing the 9+2 structure of the axoneme in the trachea, brain and oviduct, which beats in a planar back-and-forth manner (Cheung and Jahn, 1976; Katoh et al., 2018; Sasaki et al., 2019; Ueno et al., 2012; Yoke et al., 2020).

### The ciliary 3D-beat consists of fast and slow strokes, with no cessation in between

The 3D displacement of the microbeads bound near the ciliary tip was plotted against time. This analysis revealed that ciliary motion exhibited asymmetric displacements during each beat cycle (Fig. 2A), comprising fast and slow strokes. Separate plots of the cyclic *x*, *y* and *z* displacements versus time showed that whereas the positive and negative slopes along the *x*- and *z*-axes were nearly equal (though mirrored), the slopes with time along the *y*-axis differed (Fig. 2B–D), indicating that the observed asymmetry in microbead movement was mainly due to faster or slower displacements along the *y*-axis. The 3D instantaneous speed of the microbead bound to the individual cilium for every two frames showed that the repeated asymmetric beating with a CCW rotary pattern of an individual cilium consisted of two distinct phases – a faster stroke backward along a path away from the cell surface and a slower stroke forward along a path near the cell surface (Fig. 2E). Plots of the development time of the 3D instantaneous speed further confirmed this repetitive pattern, displaying alternating faster and slower speed (Fig. 2F). Using our setup, the average, fastest, and lowest 3D ciliary speeds derived from the trajectories of 20 individual cilia were 552±281 μm s$^{-1}$, 1013±433 μm s$^{-1}$, and 220 ±156 μm s$^{-1}$ (mean±s.d., *n*=20 cells), respectively (Fig. 2G). A broad distribution of ciliary beat frequencies was observed, similar to that reported recently (Soh et al., 2022). We also examined the relationship between the average 3D beating speed (*v*) and 3D beating frequency (*f* ) (Fig. 2H). When the speed was reduced, the beat frequency decreased at a fixed ratio. This linear relationship (*v*=*f*×*L*, where *L* is the trajectory length per beat cycle) suggests that the path length of a single ciliary beat is relatively robust, even when the beating speed or frequency is affected by factors, such as micropipette aspiration of the cell or biotinylation treatment of the cilia. Bead-attached cilia on the anterior side tended to beat faster during the fast stroke than those on the posterior side, although not significantly (Fig. S4), suggesting that cell-to-cell variability might underlie differences in ciliary behaviour.

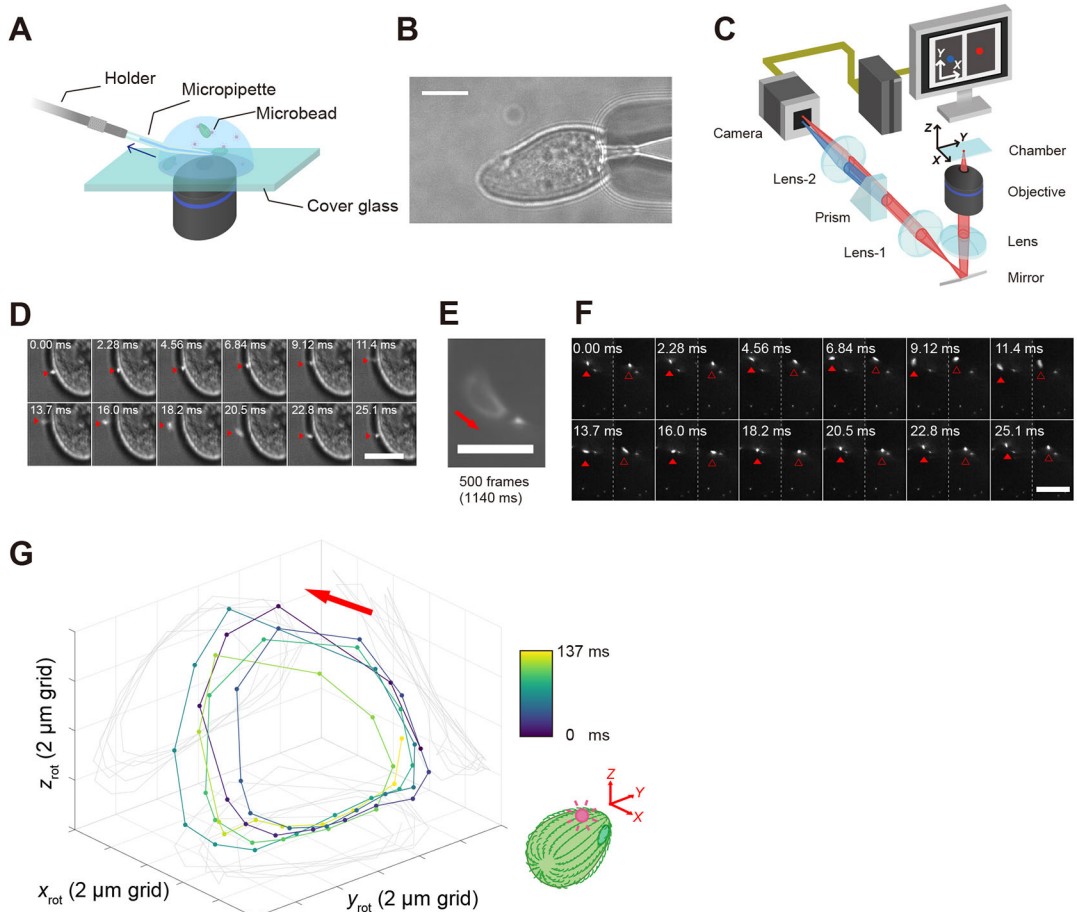

**Fig. 1. 3D tracking of the tip movement of an individual cilium of a live *Tetrahymena* cell.** (A) Schematic of the experimental setup. A *Tetrahymena* cell in which fluorescent microbeads (pink) were bound to the cilia was trapped using a micropipette. (B) Bright-field microscopy image of a *Tetrahymena* cell trapped using a micropipette. Scale bar: 20 µm. (C) Schematic of a tPOT microscope. The $z$ and $xy$ positions of a microbead are obtained from a pair of images split by the prism. (D) Sequential images of the cell and a microbead bound to the cilium during one beating cycle obtained by simultaneous bright-field and fluorescence imaging (time in milliseconds). The red arrowheads indicate the position of the microbead. Scale bar: 15 µm. (E) Superimposing successive images of a microbead bound to the cilium for 1.14 s (~42 beating cycles). The red arrow indicates the beat direction. Scale bar: 15 µm. (F) Sequential images of the microbead bound to the cilium during one beating cycle. The filled and open arrowheads indicate a pair of images of a single microbead, split by a prism (time in milliseconds). Scale bar: 20 µm. Data were obtained from the observation performed in E using a tPOT microscope at 438 frame $s^{-1}$. (G) 3D trajectory of the microbead bound to the cilium (~5 beating cycles; full tracking data available in Fig. S3A). The original trajectory data were rotated such that the $z_{rot}$-axis was approximately perpendicular to the cell surface from which the cilium with the attached microbead protruded, the $x_{rot}$-axis aligned with the left-right axis of the cell, and the cell was oriented along the positive $y_{rot}$-axis, corresponding to the swimming direction. The location of the oral apparatus was not identified but is depicted in the schematic to indicate the anterior–posterior axis for orientation in the analysis. The rotated 3D trajectory reveals the counter-clockwise rotational motion of the beating cilium viewed from the above (red arrow). Colour indicates the observation time (see the colour bar). Data in the figure are representative of seven independent experiments, each performed using separately cultured cells.

The instantaneous 3D speed of the beating cilia showed no clear significant decrease in speed or pause at the point where the stroke direction against the cell longitudinal axis switched (Fig. 2E,F), indicating that the ciliary tip made a smooth gyrating motion with periods of slow and fast movement instead of resting. This differs from the beating of tracheal cilia, which also contain a 9+2 microtubule structure with a rest phase at the end of the effective stroke (Ueno et al., 2012). Although we cannot exclude the possibility that the rotary motion of the ciliary tip was caused by trapping the cell using a micropipette, the observed beat frequency was comparable to values reported in the literature (Soh et al., 2022; Wood et al., 2007); hence, we cannot say that this is necessarily an artefact resulting from trapping by the micropipette. Rather, it is only by 3D tracking of individual ciliary tips in live ciliates that the semi-circular path of the ciliary tip during a stroke could be detected. A recent report demonstrated that motile cilia lacking

radial spoke head components in the trachea, ependymal tissues and oviduct exhibit beating with a rotary motion instead of beating in the back-and-forth motion that wild-type motile cilia perform (Shinohara et al., 2015; Yoke et al., 2020). Radial spoke-dependent dynein activity in the axoneme might regulate a variety of ciliary beating motions, including rotary motion (Lindemann and Lesich, 2010; Smith and Yang, 2004). 3D motion analysis of ciliate cilia mutated for axonemal components (Urbanska et al., 2018), including radial spokes and central apparatus, will provide more insight into the common mechanism of ciliary movement.

## Direction of the fastest motion on the beat cycle is toward the posterior left

Next, we quantified the direction of the fastest motion during ciliary beating in three dimensions, which was hypothesised to generate the propulsive force. To this end, the 3D-trajectories of the microbeads

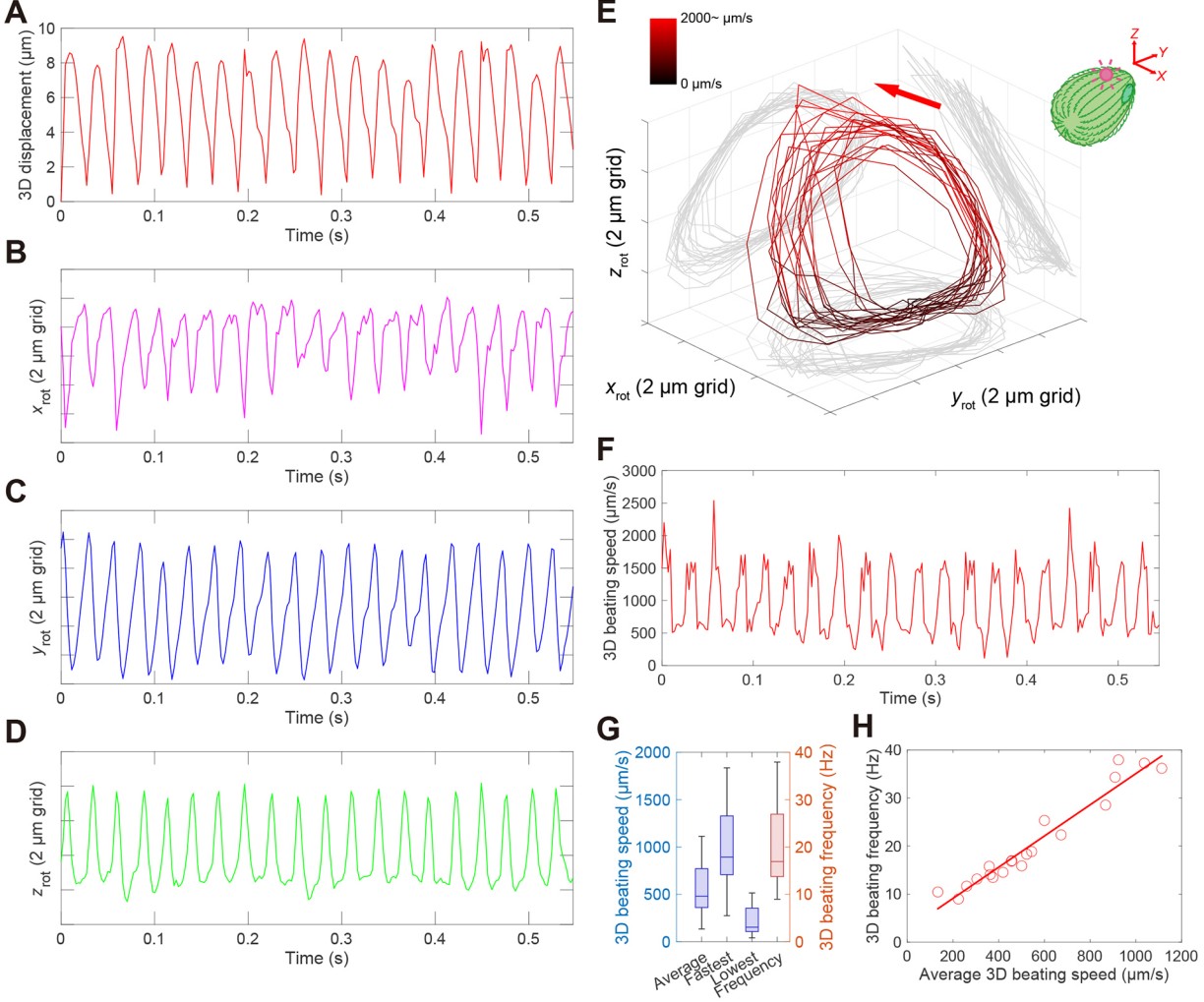

**Fig. 2. 3D beating speed of an individual cilium during the entire beating cycle.** (A–D) Time course of xyz- (A), x- (B), y- (C) and z- (D) axis motion of a microbead bound to the single cilium (~20 beating cycles; full tracking data available in Fig. S3A). (E) Rotated 3D trajectory of the microbead. The colour indicates instantaneous 3D beating speeds (see the colour bar), which were calculated from the 3D-displacement difference of a microbead between successive frames divided by the time interval. (F) Time course of instantaneous 3D beating speed of the microbead shown in E. Data in A–F are representative of seven independent experiments, each performed using separately cultured cells. (G) Box plot of average, fastest and lowest speeds of ciliary 3D motion (left, blue) and 3D beating frequency (right, red). The box represents 25th and 75th percentiles, and median is indicated as blue line in the box (*n*=20). (H) Correlation between 3D beating speed and beat frequency. The correlation coefficient was 0.97, suggesting a strong correlation between the two variables. Average ciliary beat frequency was 21±9 Hz (mean±s.d., *n*=20).

bound to a single cilium from one individual cell was fitted to a single plane and divided into 36 sections with polar coordinates whose origin was the centroid with declination every 10°. The beating speed was then estimated for each section based on the transit time (Fig. 3A). When the fast and slow movements of the ciliary tip were indicated by a bright red and dark red line segment, corresponding to the effective and recovery strokes, respectively, the overall trajectory appeared semi-circular, with the effective stroke tracing an arc and the recovery stroke tracing a near-linear segment of the path. We also performed phase-averaged all available data from different cells to derive a representative beat cycle and velocity profile (Fig. S5), which confirmed consistency with Fig. 3A. The average direction and speed of the fastest movements of the microbeads bound to each cilium in three dimensions are shown in Fig. 3B. The mean fastest motion of the ciliary tip was toward the posterior left, almost parallel to the cell surface, but slightly away from it (Fig. 3B).

Our quantification revealed that during fast strokes, the ciliary tip follows a circular trajectory directed away from the cell surface,

whereas during slow strokes, it traces a near-linear path close to the cell surface (Fig. 3A and Fig. S5). Although our observation technique cannot detect the full 3D shape of an individual cilium, the 3D trajectories of the ciliary tip, together with the 2D shape of the cell and the cilium obtained from the bright-field images at a single focal plane (Fig. S6; Movie 3), suggest a simple whip-like motion of the cilium, with its tip following a semi-circular trajectory on approximately the same plane in a 3D space (Fig. 3C). From these trajectories, we estimated that the direction of the effective stroke points slightly left and backward relative to the cell body (Fig. 3B). According to theoretical predictions from the helix theorem in microorganism locomotion (Cicconofri and DeSimone, 2019; Rossi et al., 2017), the asymmetry between effective and recovery strokes generates incremental translations and rotations which, when repeated over time, give rise to helical swimming paths (Crenshaw, 1996; Crenshaw and Edelstein-keshet, 1993; Rossi et al., 2017). Within this framework, such asymmetric strokes executed in a coordinated fashion across the ciliary array can

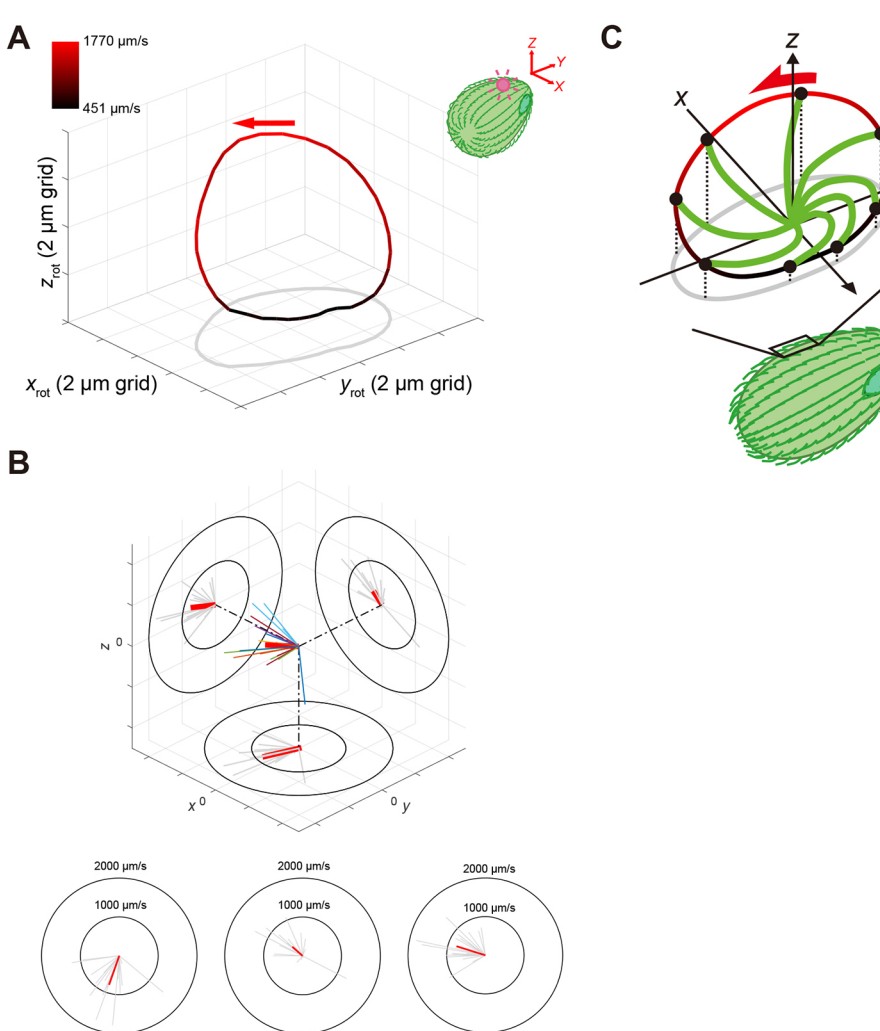

**Fig. 3. The mean direction of the fastest stroke of ciliary tip in three dimensions.** (A) Rotated average 3D trajectory obtained from ~20 beating cycles of the microbead bound near the tip of an individual cilium. Data were obtained from Fig. S3A. The colour indicates the average speed (see the colour bar). The red arrow indicates the direction in which the microbead moves at the average fastest speed in the section divided into 36. (B) 3D direction of the mean fastest speed of moving microbeads bound to cilium. In a 3D space, each colour of the line represents the direction of the average fastest speed of microbeads bound to different cilia (*n*=20 cilia on 20 cells). Each average direction was projected onto the *x-y*, *x-z* and *y-z* planes. The length of each red line indicates the average fastest speed of individual cilia. Thick red lines indicate the mean direction and speed of the average fastest strokes. (C) 3D trajectory of the ciliary tip during a beat cycle traces a semi-circular CCW motion (tip-to-base view), with the effective and recovery strokes represented by a bright red arc and a dark red line. The fastest stroke is directed from right-forward to left-back (red arrow). The ciliary shapes (green line) with varying curvatures are inferred from 2D side-view images (effective stroke) or speculated (recovery stroke) (Fig. S6). Data in the figure are representative of seven independent experiments, each performed using separately cultured cells.

generate both forward propulsion and body rotation. Our finding thus provides a plausible biomechanical basis for the right-handed helical swimming trajectory previously reported in *Tetrahymena* (Marumo et al., 2021). The slightly left-backward direction of the effective stroke is consistent with a net forward movement accompanied by rightward body rolling, as predicted by the helix theorem. This interpretation supports classical models of ciliary coordination underlying helical swimming, such as *Paramecium* (Eckert, 1972; Machemer and Sugino, 1989).

Recent studies have demonstrated that another protist, *Chlamydomonas reinhardtii*, achieves controlled helical navigation through inherently 3D, non-planar flagellar beating patterns (Striegler et al., 2025), revealing the direct link between beat geometry and trajectory curvature (Cortese and Wan, 2021). Furthermore, Sartori et al. (2016) showed that dynamic curvature regulation in the axoneme accounts for both symmetric and asymmetric (non-planar) flagellar waveforms, which in turn generate the helical swimming paths observed in *Chlamydomonas*. These findings strongly support the view that non-planar ciliary or flagellar beating is a fundamental mechanism underlying helical swimming trajectories in microorganisms. 3D observation of the movement of individual cilium on freely swimming cells could provide further insight into the dynamic coordination of beat patterns responsible for directional control and adaptive motility.

## MATERIALS AND METHODS
### Preparation of avidin-coated microbeads and biotinylated cells
Carboxylated fluorescent microbeads [0.2 µm in diameter, F8810, red fluorescent (580/605), Thermo Fisher Scientific] were crosslinked with neutravidin (0.5 mg ml$^{-1}$, Wako Pure Chemical, Osaka, Japan). Microbeads were initially pelleted and resuspended in the activation buffer (100 mM MES, pH 6.0). Carboxyl groups on the surface of microbeads were functionalised with amine reactive groups via 1-ethyl-3-(3-dimethylaminopropyl) carbodiimide (Thermo Fisher Scientific) and sulfo-hydroxysuccinimide (Thermo Fisher Scientific) crosslinking at 23°C for 15 min. The microbeads were washed with H-buffer (10 mM tricine, 0.5 mM MOPS, 8 mM NaCl, 50 µM CaCl$_2$, pH 8.4) and then neutravidins were added to the microbead and reacted at 23°C for 2 h. Excess neutravidins were removed by centrifugation (14,400 *g* for 3 min, repeated four times). The microbeads were resuspended in M-buffer (10 mM tricine, 0.5 mM MOPS, 8 mM NaCl, 50 µM CaCl$_2$, pH 7.4). The microbeads (~1 nM) were stored at 4°C and used within 2 weeks. *Tetrahymena thermophila* cells (SB255, kindly provided by Professor Y. Toyoshima of the University of Tokyo) were initially pelleted and resuspended in H-buffer and then biotinylated with biotin-(AC$_5$)$_2$-Sulfo-OSu (Dojindo, Kumamoto, Japan) at 37°C for 30 min. Excess biotin was removed by centrifugation (400 *g* for 1 min, repeated six times). The cells were resuspended in M-buffer and observed within 3 hours.

### Trapping of *Tetrahymena* with microbead-bound cilia
10 µl of biotinylated *Tetrahymena* in M-buffer were put onto a coverglass (NEO Micro cover glass; thickness no. 1; 24×36 mm, Matsunami Glass,

Journal of Cell Science

Osaka, Japan) and then 10 μl of neutravidin-coated microbeads (∼0.01 nM) were added. A glass micropipette (30 μm OD, 5 μm ID and the 20-degree curved tip, L-Tip, Yodaka, Kanagawa, Japan) was used to trap *Tetrahymena* with microbead-bound cilia. Manipulation of the micropipette was done using a micromanipulator (MHW-3, Narishige, Tokyo, Japan), and pressure was controlled using a microinjector (IM-9B, Narishige, Tokyo, Japan).

## Observation of the 3D motion of a microbead attached to the cilium

A microbead bound to an individual cilium of live *Tetrahymena* trapped using a micropipette were observed under the tPOT microscope (Yajima et al., 2008). Assays were carried out at 23±1°C. The tPOT microscope, which provides *z*-positional information from planar images with nanometre accuracy, uses a prism to split one image into two and calculates *z*-positional information of the sample from the difference in *y*-positions of the two images. As illustrated in Fig. 1C, the back-focal-plane (BFP) of the objective (UPLSAPO60XW, 60×, NA 1.2, Olympus) was focused outside the camera port of an inverted microscope (IX70, Olympus) with achromatic Lens-1 (combined focal length 170 mm) to make an equivalent BFP (eBFP). To split the image beam path at the eBFP, a custom-made wedge prism (91.0°, Natsume Optics, Nagano, Japan) coated with an antireflective layer was precisely located at the eBFP. The two split images of a sample were focused on the camera focal plane by achromatic Lens-2 (combined focal length 170 mm). Images were recorded by CMOS camera (C14440-20UP, Hamamatsu Photonics, Shizuoka, Japan) at 2.28 ms per frame (2×2 binning).

## Data analysis

The positions of the two optically separated images of a fluorescent microbead were determined by 2D Gaussian fitting as $(x_1, y_1)$ and $(x_2, y_2)$, and *x*, *y* and *z* were calculated as $(x_1+x_2)/2$, $(y_1+y_2)/2$ and $(y_1−y_2)/2$, respectively. For calibration of *z*-axis real position and $(y_1−y_2)/2$, a custom-built stable stage (Chuukousha Seisakujo, Tokyo, Japan) equipped with a pulse motor (SGSP-13ACTR-BO, Sigma Koki, Tokyo, Japan) and controller (QT-CM2, Chuo Precision Industrial, Tokyo, Japan) was used to move the objective vertically placed to the stage while observing a stable fluorescent microbead (0.2 μm in diameter, F8810, Thermo Fisher Scientific) placed inside the observation chamber filled with 1.0% agarose gel (Agarose S, Nippon Gene, Tokyo, Japan). The calculated *z*-position and actual *z*-position (as defined by the pulse motor) corresponded linearly over a range of ±2.5 μm from the focal plane. The 3D-trajectories of the microbeads were obtained using a custom-written software (Mat3) in MATLAB (MathWorks) (available upon request). The instantaneous 3D beating speeds were calculated based on the 3D-displacement difference of a microbead between successive frames divided by the time interval. For the analysis of the average speed and the fastest stroke direction, the 3D trajectories of the microbeads were fitted to a single plane in 3D space using principal component analysis (PCA, MATLAB) and then projected onto it. The projected trajectories were divided into 36 angular sections, with the centroid as the origin and an angular increment of 10° per section. Subsequently, the inter-section average speed was estimated for each section based on the transit time of each section. From this averaged trajectory, the fastest and slowest speeds were determined as the maximum and minimum inter-section speeds, respectively. The average speed was calculated as the weighted arithmetic mean of all inter-section speeds, with the weights corresponding to the number of data points within each section. Data are provided in Table S1.

## Acknowledgements
We thank Mitsuhiro Sugawa and Yoko Y. Toyoshima for critical discussion.

## Competing interests
The authors declare no competing or financial interests.

## Author contributions
Conceptualization: J.Y.; Data curation: A.M., H.I., S.Y., R.S.; Formal analysis: A.M., H.I.; Funding acquisition: M.Y., H.I., J.Y.; Investigation: A.M., H.I., S.Y., R.S.; Methodology: A.M., H.I., S.Y., K.M., M.Y.; Software: A.M.; Supervision: M.Y., J.Y.;

Validation: A.M., H.I., R.S.; Visualization: A.M., S.Y., H.I., R.S.; Writing – original draft: J.Y., A.M., H.I., M.Y.; Writing – review & editing: J.Y., A.M., H.I., M.Y.

## Funding
This work was supported in part by Japan Society for the Promotion of Science KAKENHI (grant numbers 25K02235, 21K19252 to J.Y. and grant number 23K14177 to M.Y.); the Ministry of Education, Culture, Sports, Science and Technology (MEXT) KAKENHI Grant-in-Aid for Transformative Research Areas (grant number JP23H04401 to J.Y.); Japan Science and Technology Agency (JST) SPRING (grant number JPMJSP2108 to H.I.); the Precise Measurement Technology Promotion Foundation (PMTP-F to J.Y.) and Uehara Memorial Foundation to J.Y. Open Access funding provided by The University of Tokyo. Deposited in PMC for immediate release.

## Data and resource availability
The datasets generated and/or analysed, and all samples used in this study, as well as custom scripts for tracking beads and analysis of trajectories are available from the corresponding author on reasonable request. All relevant data can be found within the article and its supplementary information.

## Peer review history
The peer review history is available online at https://journals.biologists.com/jcs/lookup/doi/10.1242/jcs.264027.reviewer-comments.pdf

## Special Issue
This article is part of the Special Issue 'Cilia and Flagella: from Basic Biology to Disease', guest edited by Pleasantine Mill and Lotte Pedersen. See related articles at https://journals.biologists.com/jcs/issue/138/20.

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
