## [Peer Review File · Journal of Cell Science]

Three-dimensional beating pattern of ciliary tip in the live ciliate *Tetrahymena*

Akisato Marumo, Hiroto Ishii, Shin Yamaguchi, Rieko Sumiyoshi, Kyohei Matsuda, Masahiko Yamagishi and Junichiro Yajima
DOI: 10.1242/jcs.264027

Editor: Lotte Pedersen

Review timeline

Original submission:	26 March 2025
Editorial decision:	8 May 2025
First revision received:	10 July 2025
Accepted:	6 August 2025

Original submission

First decision letter

MS ID#: jcs.264027

MS TITLE: Three-dimensional beating pattern of individual ciliary tip in the live ciliate *Tetrahymena*

AUTHORS: Junichiro YAJIMA; Akisato Marumo; Hiroto Ishii; Shin Yamaguchi; Rieko Sumiyoshi; Kyohei Matsuda; Masahiko Yamagishi

ARTICLE TYPE: Short Report

Dear Dr YAJIMA,

We have now reached a decision on the above manuscript.

To see the reviewers' reports and a copy of this decision letter, please go to:

Reviewer 1

Advance summary and potential significance to field

In this short report, the authors describe the system they have developed for observing the three-dimensional beating of *Tetrahymena thermophila* motile cilia. This is an important area of research as it is difficult to quantify the waveform of motile cilia beating in anything but two dimensions. The authors do a great job of using their technology to track the movement of the tip of motile cilia and even though they did not image the cilium itself, having the three dimensional path of movement gives new information about the waveform of ciliary beating. The authors are also able to use their results to explain the right-handed helical swim pattern of *Tetrahymena* and shed light on how subtle directional variation of ciliary movement can result in large scale directional outputs. This type of information is of broad interest to ciliary biologists trying to determine the subtle contributions of the many molecular components of the ciliary axoneme as well as to the computational modeling community that will now have new inputs to refine models of microorganism swimming and directed fluid flow. I fully support the publication of this work if the authors can address the few minor concerns I have.

Comments for the author

Minor critiques:

The authors are creative in how they present the data but I think a small change for clarity would help the reader. In Figure 1G, reducing the timescale to reduce the number of beat cycles would allow the reader to focus their attention on the movement more easily. I have the same issue with Figure 2E. At some level after a few beat cycles the data is redundant, unless the variability between beat cycles is the point that is trying to be made.

It would be interesting to hear commentary on the variability between the beat trajectory of multiple different cilia since most everything represented is a single cilium displayed over many beat cycles. Also, it would be informative to know where along the body axis of the cell are the cilia being imaged from. In *Tetrahymena*, cilia at different regions (anterior, medial, posterior) of the cell beat with different periodicities and this may account for the discrepancy between beat cycle timing.

It is not completely clear how the authors know that the bead is attached to the tip of the cilium and not somewhere along the side of the cilium since they are using a biotinylation method directed at cilium surface proteins. If the authors can expand on how they discerned which beads were actually at the tip versus the sides of the cilium that would help interpret the data they show and could explain any variability in data.

Minor text change suggestions:

The species of *Tetrahymena* should be named in the manuscript the first time that *Tetrahymena* is mentioned. It is only at line 78 that it is revealed that the authors were working with *Tetrahymena thermophila* even though *Tetrahymena* had been referenced numerous times before then. I mention this because new research showing the swim speed of *Tetrahymena* varies across species is emerging.

Reviewer 2*Advance summary and potential significance to field*

This paper reports measurements of the 3D tip trajectory of *Tetrahymena* cilia, by monitoring the orbit of fluorescent microspheres bound to cilia using a tPOT tracking microscope. The main result concerns the shape of the tip movement - which is shown to follow stereotypical semi-circular orbits in 3D.

Such three-dimensional beat patterns are likely essential for generating helical swimming motion in ciliated and flagellated swimmers. The scope of the work is suitable in principle for *Journal of Cell Science*, but I have some concerns about this paper before it can be accepted, outlined below.

Comments for the author

1. Novelty - there was a similar paper by the same (or similar group of) authors in *communications biology* a few years ago (Marumo et al, 2021, *Commun Biol*), 'Three dimensional tracking of the ciliate *Tetrahymena* reveals the mechanism of ciliary stroke-driven helical swimming', please clarify how results here are different? What is the innovation in this new work, what are the new insights since the previous work? It's important to place the previous literature in context.

2. Related to this, it seems to me that one key distinction compared to the 2021 paper is that here the measurements were performed here for a micropipette-fixed cell, correct? So rather than just stating 'we cannot exclude the possibility that the rotary motion of the ciliary tip was caused by trapping the cell using a micropipette' - logically there should be a way to compare the 3D tip trajectories measured in the current work with those measured previously for a freely swimming cell? Presumably you have access to both sets of data? That way you can be sure if the pipette,

which adds an external force not present in the free-swimming problem, actually makes any difference, this would also be interesting and important to know.

3. There was a tendency to cite the older literature - this itself is not a problem, but the field has made a lot of progress since then and for completeness newer results should be acknowledged too. For example in the section Direction of the fastest motion... 'the results also support the model of ciliary beating control in the helical swimming pattern of paramecium'. Eckert 1972 and Machemer 1989 were cited, but several much more recent papers on 3D movement of microorganisms have not been discussed. Recent studies (both experimental and computational) have explicitly related the helical trajectories of certain species of microswimmers (for example *Chlamydomonas*) to the observed 3D beat pattern of the cilia.

4. What is the significance of the measured linear scaling between frequency and beat speed in figure 2H? were the data points all from different individuals? Or could it be that there is some gradient in beat frequency across the organism? Does the method allow for discrimination of the position of the tracked cilium relative to the anatomy of the cell? If so, could the analysis be repeated to deduce if there are any interesting spatial patterns across the organism?

5. What kind of data averaging was performed? What about a phase averaging to get the stereotypical shape of a beat cycle not just averaged across one cilium over many cycles as in 3A but over all the available data across different individuals? This dataset (including coordinates) could then be uploaded to a repository or as Supplementary information for others to access?

6. Quality of figures not great - for example figure 2 panels - some axis labels are barely visible. Figure 1G - the two panels are just rotated versions of each other? Seems redundant.

7. I would suggest some rewriting to highlight the implications of this result, beyond rather vague statements about how the beat patterns may or may not be related to 3D helical swimming. The relationship has been made concrete in several previous papers, for example following arguments similar to the Cicconofri & De Simone paper, and other work (see comment 3).

First revision

Author response to reviewers' comments

Reviewer 1: SUMMARY OF THE ADVANCE MADE IN THIS PAPER AND ITS POTENTIAL SIGNIFICANCE TO THE FIELD

*In this short report, the authors describe the system they have developed for observing the three-dimensional beating of *Tetrahymena thermophila* motile cilia. This is an important area of research as it is difficult to quantify the waveform of motile cilia beating in anything but two dimensions. The authors do a great job of using their technology to track the movement of the tip of motile cilia and even though they did not image the cilium itself, having the three dimensional path of movement gives new information about the waveform of ciliary beating. The authors are also able to use their results to explain the right-handed helical swim pattern of *Tetrahymena* and shed light on how subtle directional variation of ciliary movement can result in large scale directional outputs. This type of information is of broad interest to ciliary biologists trying to determine the subtle contributions of the many molecular components of the ciliary axoneme as well as to the computational modeling community that will now have new inputs to refine models of microorganism swimming and directed fluid flow. I fully support the publication of this work if the authors can address the few minor concerns I have.*

We thank the reviewer for the positive evaluation of our work, and for the constructive comments and suggestions. Please find our point-by-point responses to the reviewer's queries below.

SUGGESTIONS TO AUTHORS

Minor critiques:

1) The authors are creative in how they present the data but I think a small change for clarity would help the reader. In Figure 1G, reducing the timescale to reduce the number of beat cycles would allow the reader to focus their attention on the movement more easily. I have the same issue with Figure 2E. At some level after a few beat cycles the data is redundant, unless the variability between beat cycles is the point that is trying to be made.

Following the reviewer's suggestion, we revised Figures 1G and 2E by decreasing the number of beat cycles in order to more clearly present the trajectory of the beating motion. In line with this revision, we also narrowed the displayed time window in Figures 2A-D and F. The original graphs showing the full trajectories are provided in Supplementary Figure 3A.

2) It would be interesting to hear commentary on the variability between the beat trajectory of multiple different cilia since most everything represented is a single cilium displayed over many beat cycles.

Thank you for your thoughtful comments. As noted, our imaging system currently makes it challenging to analyse the 3D positions of beads when multiple cilia are labelled in close proximity, due to image overlap. In addition, high-speed imaging necessitates limiting the field of view in order to maintain sufficient temporal resolution. As a result, it is not feasible to simultaneously observe beads bound to cilia that are far apart within the same cell. For these reasons, we conduct our experiments under conditions in which typically only a single bead binds to one cilium per cell, and the vast majority of cilia remain unlabelled. Whilst we have explored conditions to label multiple, spatially separated cilia within the same focal volume, we have not yet succeeded in establishing such conditions with sufficient resolution for reliable 3D analysis.

Also, it would be informative to know where along the body axis of the cell are the cilia being imaged from. In Tetrahymena, cilia at different regions (anterior, medial, posterior) of the cell beat with different periodicities and this may account for the discrepancy between beat cycle timing.

In addition, the cilium itself and its base are not clearly visible in our current imaging setup, which makes it difficult to precisely determine the position of the bead-labelled cilium on the cell surface. However, by combining the fluorescent signal from the bead with the overall cell morphology in the bright-field image, we were able to roughly classify the location of the cilium along the cell body axis as either anterior or posterior. Representative examples of the approximate bead binding positions on individual cells are illustrated in the schematic diagrams of *Tetrahymena* shown in Figure 1G and supplementary Figure 3B-F. Based on this classification, we created box plots of 3D beating speeds (for average, fast, and slow strokes) from 20 individual cells, categorised by anterior and posterior cilium positions. These results have been newly included as Supplementary Figure 4. Although no statistically significant differences were observed in our measurement, the data show a tendency for anterior cilia to exhibit higher average 3D beating speeds during fast strokes. We have added this point to the revised manuscript.

Page 8, lines 154-156

Bead-attached cilia on the anterior side tended to beat faster during the fast stroke than those on the posterior side, though not significantly (Supplementary Fig. 4), suggesting that cell-to-cell variability may underlie differences in ciliary behaviour.

3) It is not completely clear how the authors know that the bead is attached to the tip of the cilium and not somewhere along the side of the cilium since they are using a biotinylation method directed at cilium surface proteins. If the authors can expand on how they discerned which beads were actually at the tip versus the sides of the cilium that would help interpret the data they show and could explain any variability in data.

We appreciate this insightful comment. We agree that clarifying the bead attachment position is

crucial for accurately interpreting the 3D trajectory data. To address this issue, we immobilized cells with bead-attached cilia, prepared using a method equivalent to that in the main experiments, on glass substrates with 2% paraformaldehyde, and observed the bead attachment positions on the cilia by confocal microscopy. The results from $n = 20$ cells showed that the fluorescent beads were consistently located at or near the distal tips of the cilia, rather than along their lateral surfaces, suggesting that the beads preferentially bind to the ciliary tips. Representative images are included in the revised Supplementary Figure 1.

Minor text change suggestions:

1) The species of Tetrahymena should be named in the manuscript the first time that Tetrahymena is mentioned. It is only at line 78 that it is revealed that the authors were working with Tetrahymena thermophila even though Tetrahymena had been referenced numerous times before then. I mention this because new research showing the swim speed of Tetrahymena varies across species is emerging.

Thank you for pointing that out. We have now specified the species name we used, *Tetrahymena thermophila*, at its first mention in the Abstract as recommended.

2) Supplemental Figure 3 legend. First sentence, change “One beating of the cilium” to “one beat cycle of the cilium.”

Thank you for the suggestion. We have revised the first sentence to read “one beat cycle of the cilium” as recommended.

Reviewer 2: SUMMARY OF THE ADVANCE MADE IN THIS PAPER AND ITS POTENTIAL SIGNIFICANCE TO THE FIELD

This paper reports measurements of the 3D tip trajectory of Tetrahymena cilia, by monitoring the orbit of fluorescent microspheres bound to cilia using a tPOT tracking microscope. The main result concerns the shape of the tip movement - which is shown to follow stereotypical semi-circular orbits in 3D. Such three-dimensional beat patterns are likely essential for generating helical swimming motion in ciliated and flagellated swimmers. The scope of the work is suitable in principle for Journal of Cell Science, but I have some concerns about this paper before it can be accepted, outlined below.

We appreciate the reviewer’s thorough review of our manuscript and their constructive criticisms and suggestions. We are especially grateful to hear that our manuscript is, in principle, considered suitable for publication in Journal of Cell Science. Please find our point-by-point responses to the reviewer’s queries below.

SUGGESTIONS TO AUTHORS

1. Novelty - there was a similar paper by the same (or similar group of) authors in communications biology a few years ago (Marumo et al, 2021, Commun Biol), Three dimensional tracking of the ciliate Tetrahymena reveals the mechanism of ciliary stroke-driven helical swimming’, please clarify how results here are different? What is the innovation in this new work, what are the new insights since the previous work? It’s important to place the previous literature in context.

We thank the reviewer for raising this important point. Indeed, the study by Marumo et al. (2021) in *Communications Biology* provided valuable insights into the 3D helical swimming behaviour of ciliate *Tetrahymena*. Our current work builds upon and extends these findings in several ways:

Detailed Analysis of Individual Ciliary Beat Cycles: While the previous study focused on the overall 3D swimming trajectory of *Tetrahymena* cell by tracking cells that had phagocytosed microbeads as markers (Marumo et al. *Communications Biology* 2021), our current work investigates the 3D beating trajectories of individual ciliary tips over multiple beat cycles by attaching microbeads to the cilium. This approach reveals finer details of 3D ciliary motion

dynamics that were not previously reported in ciliate *Tetrahymena*.

Higher Spatial Resolution: We applied a three-dimensional prismatic optical tracking (tPOT) microscope (Yajima et al. *Nature Structural Molecular Biology* 2008), originally developed by our group for studying molecular motor proteins and cells, to visualise ciliary tip motion in 3D. This approach enabled us to capture more detailed and dynamic aspects of *Tetrahymena* ciliary beating that were not accessible in previous studies.

These advances provide new insights into the mechanisms of ciliary motion and their contribution to cell swimming behaviour. We have revised the manuscript accordingly to clarify these points and to better contextualise our findings within the existing literature.

Pages 4-5, lines 75-83,

Recently, we reported that *Tetrahymena* swims along a right-handed helical trajectory at a high speed of approximately $400 \mu\text{m s}^{-1}$, accompanied by right-handed rolling of its cell body in free space (Marumo et al., 2021). This finding was obtained through 3D tracking of *Tetrahymena* cells that had internalised fluorescent microbeads via phagocytosis, using a three-dimensional prismatic optical tracking (tPOT) microscope originally developed by our group (Yajima et al., 2008). Due to the dense protrusion of numerous cilia from the surface of *Tetrahymena* (Bayless et al., 2019), which swims rapidly in a helical manner (Ishikawa and Kikuchi, 2018; Marumo et al., 2021), it is difficult to observe the 3D motion of individual cilia in live *Tetrahymena*. In this study, we tracked the 3D trajectories of fluorescent microbeads attached to the tips of cilia in live, immobilised *Tetrahymena thermophila*.

2. Related to this, it seems to me that one key distinction compared to the 2021 paper is that here the measurements were performed here for a micropipette-fixed cell, correct? So rather than just stating 'we cannot exclude the possibility that the rotary motion of the ciliary tip was caused by trapping the cell using a micropipette' - logically there should be a way to compare the 3D tip trajectories measured in the current work with those measured previously for a freely swimming cell? Presumably you have access to both sets of data? That way you can be sure if the pipette, which adds an external force not present in the free-swimming problem, actually makes any difference, this would also be interesting and important to know.

Thank you for your comment. We would like to clarify that our previous paper (Marumo et al., *Communications Biology* 2021) focused solely on the observation of cell swimming trajectories in 3D and did not include direct measurements or analysis of individual ciliary motion. Therefore, there is no corresponding dataset of ciliary tip trajectories from freely swimming cells available for direct comparison with the current micropipette-fixed measurements. We have now clarified this distinction in the revised manuscript to avoid potential confusion. Since this point is also related to our response to Comment 1 from this reviewer, some of the revised sections overlap. Please refer to our response to Comment 1 for details.

3. There was a tendency to cite the older literature - this itself is not a problem, but the field has made a lot of progress since then and for completeness newer results should be acknowledged too. For example in the section Direction of the fastest motion... the results also support the model of ciliary beating control in the helical swimming pattern of paramecium'. Eckert 1972 and Machemer 1989 were cited, but several much more recent papers on 3D movement of microorganisms have not been discussed. Recent studies (both experimental and computational) have explicitly related the helical trajectories of certain species of microswimmers (for example *Chlamydomonas*) to the observed 3D beat pattern of the cilia.

We appreciate the reviewer for pointing out the importance of including more recent studies on the 3D movement of microorganisms and their ciliary beat patterns. As suggested, we have revised the relevant section to incorporate and discuss recent findings, particularly from studies on *Chlamydomonas*, which have explicitly linked the helical swimming trajectories to the three-dimensional nature of flagellar beating. We have added a discussion of recent studies for *Chlamydomonas* (Cortese & Wan, *Phys. Rev. Lett.*, 2021, Striegler et al., *Nat Phys.* 2025, Sartori et al. *eLife*, 2016) in the revised manuscript to better place our results in the context of the broader literature.

Pages 10-11, lines 210-217

Recent studies have demonstrated that another protist *Chlamydomonas reinhardtii* achieves controlled helical navigation through inherently 3D, non-planar flagellar beating patterns (Striegler et al., 2025), revealing the direct link between beat geometry and trajectory curvature (Cortese and Wan, 2021).

Furthermore, Sartori et al. (Sartori et al., 2016) showed that dynamic curvature regulation in the axoneme accounts for both symmetric and asymmetric (non-planar) flagellar waveforms, which in turn generate the helical swimming paths observed in *Chlamydomonas*. These findings strongly support the view that non-planar ciliary or flagellar beating is a fundamental mechanism underlying helical swimming trajectories in microorganisms.

4. What is the significance of the measured linear scaling between frequency and beat speed in figure 2H? were the data points all from different individuals? Or could it be that there is some gradient in beat frequency across the organism? Does the method allow for discrimination of the position of the tracked cilium relative to the anatomy of the cell? If so, could the analysis be repeated to deduce if there are any interesting spatial patterns across the organism?

The data points in Figure 2H were obtained from different individual cells. It remains unclear whether the observed variability in frequency and beat speed arises from inherent differences between individual cells, or from potential damage caused by biotinylation treatment of the cell or micropipette suction applied to cell during manipulation. However, the linear relationship observed between frequency and beat speed ($v = f \times L$, where f is frequency, v is beat speed, and L is the trajectory length per beat cycle) suggests that, regardless of the cause of the variation in beat speed, the trajectory length of the bead per cycle remains largely consistent. This implies that the ciliary waveform is maintained to some extent, indicating a certain robustness in the ciliary beating pattern. In response to the reviewer's comment, we have added these points to the revised manuscript.

In our imaging system, the base of the cilium and the cilium itself cannot be clearly visualised, so the position of the bead-attached cilium can only be roughly classified as being located at either the anterior or posterior part of the cell based on the overall shape of the cell. Using this classification, we analysed 3D beating speeds during average, fast, and slow strokes, and the results are presented in Supplementary Figure 3. Although no statistically significant differences were detected in our system, anterior cilia showed a tendency toward higher average 3D beating speeds during fast strokes. We have added this point to the revised manuscript.

Page 9, lines 150-155

This linear relationship ($v = f \times L$, where L is the trajectory length per beat cycle) suggests that the path length of a single ciliary beat is relatively robust, even when the beating speed or frequency is affected by factors such as micropipette aspiration of the cell or biotinylation treatment of the cilia. Bead-attached cilia on the anterior side tended to beat faster during the fast stroke than those on the posterior side, though not significantly (Supplementary Fig. 4), suggesting that cell-to-cell variability may underlie differences in ciliary behaviour.

5. What kind of data averaging was performed?

In response to the reviewer's question regarding data averaging, we have now included a detailed description of the averaging procedure in the Methods section.

Page 13, lines 273-281

For the analysis of the average speed and the fastest stroke direction, the 3D trajectories of the microbeads were fitted to a single plane in 3D space using principal component analysis (PCA, MATLAB) and then projected onto it. The projected trajectories were divided into 36 angular sections, with the centroid as the origin and an angular increment of 10° per section. Subsequently, the inter-section average speed was estimated for each section based on the transit time of each section. From this averaged trajectory, the fastest and slowest speeds were determined as the maximum and minimum inter-section speeds, respectively. The average speed was calculated as the weighted arithmetic mean of all inter-section speeds, with the weights corresponding to the number of data points within each

section.

What about a phase averaging to get the stereotypical shape of a beat cycle not just averaged across one cilium over many cycles as in 3A but over all the available data across different individuals?

In response to the reviewer's suggestion, we performed phase averaging across all available data (i.e. from Figure 3B, n = 20 cells) from different individuals to obtain a representative shape of a beat cycle. We have added the phase-averaged trajectory and the corresponding velocity profile as Supplementary Figure 5. In the process of addressing this comment, we also refined the method used to approximate the ciliary trajectories onto a single plane in the xyz space during averaging. Based on this improvement, we reanalysed all datasets and updated Figure 3 accordingly.

Page 9, lines 183-185

We also performed phase-averaged all available data from different individuals to derive a representative beat cycle and velocity profile (Fig. S5), which confirmed consistency with Figure 3A.

This dataset (including coordinates) could then be uploaded to a repository or as Supplementary information for others to access?

As suggested, we have included this dataset (including coordinates) as Supplementary Data for others to access.

6. Quality of figures not great - for example figure 2 panels - some axis labels are barely visible. Figure 1G - the two panels are just rotated versions of each other? Seems redundant.

Thank you for pointing that out. In response to the reviewer's comment, we have revised the axis labels to improve their readability and have removed the plot showing the bead trajectory prior to the rotation process.

7. I would suggest some rewriting to highlight the implications of this result, beyond rather vague statements about how the beat patterns may or may not be related to 3D helical swimming. The relationship has been made concrete in several previous papers, for example following arguments similar to the Cicconofri & De Simone paper, and other work (see comment 3).

We appreciate the reviewer's valuable suggestion to clarify and strengthen the discussion on the implications of our findings regarding the relationship between ciliary beat patterns and 3D helical swimming. Following the reviewer's advice, we have revised the relevant section to more explicitly connect our observations to established theoretical frameworks, including those presented by Cicconofri & DeSimone and other key studies. This revised discussion emphasises how our results concretely support models of helical swimming driven by coordinated ciliary motion. The updated text can be found in the revised manuscript.

Page 10, lines 197-208

According to theoretical predictions from the Helix Theorem in microorganism locomotion (Cicconofri and DeSimone, 2019; Rossi et al., 2017), the asymmetry between the effective and recovery strokes generates incremental translations and rotations which, when repeated over time, give rise to helical swimming paths (Crenshaw, 1996; Crenshaw and Edelstein-keshet, 1993; Rossi et al., 2017). Within this framework, such asymmetric strokes executed in a coordinated fashion across the ciliary array can generate both forward propulsion and body rotation. Our finding thus provides a plausible biomechanical basis for the right-handed helical swimming trajectory previously reported in *Tetrahymena* (Marumo et al., 2021). The slightly left-backward direction of the effective stroke is consistent with a net forward movement accompanied by rightward body rolling, as predicted by the Helix Theorem. This interpretation supports classical models of ciliary coordination underlying helical swimming, such as *Paramecium* (Eckert, 1972; Machemer and Sugino, 1989).

Second decision letter

MS ID#: jcs.264027R1

MS Title: Three-dimensional beating pattern of individual ciliary tip in the live ciliate Tetrahymena

Authors: Junichiro YAJIMA; Akisato Marumo; Hiroto Ishii; Shin Yamaguchi; Rieko Sumiyoshi; Kyohei Matsuda; Masahiko Yamagishi

Article Type: Short Report

Dear Dr YAJIMA,

I am happy to tell you that your manuscript has been accepted for publication in Journal of Cell Science, pending standard publication integrity checks.